# The Cell-Specific Role of SHP2 in Regulating Bone Homeostasis and Regeneration Niches

**DOI:** 10.3390/ijms24032202

**Published:** 2023-01-22

**Authors:** Jie Zhang, Chengxinyue Ye, Yufan Zhu, Jun Wang, Jin Liu

**Affiliations:** 1Laboratory for Aging Research, State Key Laboratory of Biotherapy, National Clinical Research Center for Geriatrics, West China Hospital, Sichuan University, Chengdu 610041, China; 2State Key Laboratory of Oral Diseases, National Clinical Research Center for Oral Diseases, Department of Orthodontics, West China Hospital of Stomatology, Sichuan University, Chengdu 610041, China

**Keywords:** SHP2, homeostasis, bone microenvironment, bone remodeling, SHP2 inhibitor, SHP2 agonist

## Abstract

Src homology-2 containing protein tyrosine phosphatase (SHP2), encoded by *PTPN11*, has been proven to participate in bone-related diseases, such as Noonan syndrome (NS), metachondromatosis and osteoarthritis. However, the mechanisms of SHP2 in bone remodeling and homeostasis maintenance are complex and undemonstrated. The abnormal expression of SHP2 can influence the differentiation and maturation of osteoblasts, osteoclasts and chondrocytes. Meanwhile, SHP2 mutations can act on the immune system, vasculature and nervous system, which in turn affect bone development and remodeling. Signaling pathways regulated by SHP2, such as mitogen-activated protein kinase (MAPK), Indian hedgehog (IHH) and phosphatidylinositol-4,5-bisphosphate 3-kinase (PI3K)/protein kinase B (AKT), are also involved in the proliferation, differentiation and migration of bone functioning cells. This review summarizes the recent advances of SHP2 on osteogenesis-related cells and niche cells in the bone marrow microenvironment. The phenotypic features of SHP2 conditional knockout mice and underlying mechanisms are discussed. The prospective applications of the current agonists or inhibitors that target SHP2 in bone-related diseases are also described. Full clarification of the role of SHP2 in bone remodeling will shed new light on potential treatment for bone related diseases.

## 1. Introduction

Bone is a living rigid tissue that supports posture, protects organs, aids locomotion and participates in hematopoiesis [1]. The functioning cells in bone metabolism include osteoblast lineage cells, osteoclasts and niche cells. Osteoblast lineage cells, such as mesenchymal stem cells (MSCs), osteoblasts, osteocytes and chondrocytes, directly participate in the bone formation process during bone development and repair [2,3]. Meanwhile, osteoclasts, which originate from hematopoietic cells, are responsible for bone resorption [2]. The disruption of the dynamic balance between osteoblast lineage cells and osteoclasts causes various bone diseases, such as osteopetrosis and osteoporosis. In addition, the local niche cells in the bone microenvironment, namely immune cells, the vascular endothelial cells (ECs), as well as the sensory and sympathetic nerves, have been proven to influence osteoprogenitor cells via the paracrine or autocrine pathway [4,5]. The complex cells in bone marrow act together to sustain bone homeostasis.

Src homology-2 containing protein tyrosine phosphatase (SHP2) is a ubiquitously expressed protein tyrosine phosphatases (PTPs) [6]. The somatic mutations of SHP2 are associated with severe skeletal manifestations. The SHP2 gain-of-function (GOF) or loss-of-function (LOF) mutation can cause multiple genetic diseases characterized by bone developmental malformations, such as Noonan syndrome (NS), LEOPARD syndrome and metachondromatosis [7]. It has been reported that the SHP2 GOF mutation is responsible for 40–50% of NS cases and 90% of LEOPARD syndrome cases, whereas the SHP2 LOF mutation causes more than 50% of metachondromatosis cases [8,9,10,11,12]. The underlying mechanism of SHP2 regulates the biological functions of a range of bone functioning cells by activating signaling pathways such as mitogen-activated protein kinase (MAPK), Indian hedgehog (IHH) and phosphatidylinositol-4,5-bisphosphate 3-kinase (PI3K)/protein kinase B (AKT) signaling [8,13]. In the MAPK signaling pathway, the hyperactivation of extracellular signal-regulated kinase 1 and 2 (ERK1/2) by the SHP2 mutations leads to growth retardation in NS patients [14]. Reduced ERK1/2 phosphorylation by SHP2 deletion results in delayed chondrocytes terminal differentiation and maturation [15]. Meanwhile, SHP2 has been shown to have negatively regulated IHH protein and AKT in a mouse metachondromatosis model [16]. Moreover, SHP2 participates in osteoblasts maturation and differentiation by promoting (bone morphogenetic protein 2) BMP2 and runt-related transcription factor 2 (RUNX2)/OSTERIX signaling [17], whilst inhibiting osteoclastogenesis through (signal transducer and activator of transcription 3) STAT3-mediated receptor activator of nuclear factor-kappa B ligand (RANKL) production [18].

Although extensive studies have focused on clarifying the critical role of SHP2 in bone homeostasis, to date, the cell-specific role of SHP2 in bone functioning cells is yet to be elucidated. The knockdown of SHP2 in osteochondroprogenitor cells or osteoblasts results in decreased osteogenic differentiation and bone formation capacity [17,18]. The deletion of SHP2 in osteoclasts could impair osteoclasts formation [8]. In addition to these stromal cells, which are directly involved in bone metabolism, other niche cells in the bone microenvironment could also be regulated by SHP2. The inhibition of SHP2 in macrophages and vascular ECs reduces the secretion of proinflammatory cytokines [19] and impaired angiogenesis, separately [20]. Mice with SHP2 deletion in the neural crest cells (NCCs) displayed severe craniofacial deficits [21]. These observations suggest that the role of SHP2 in bone niches is quite complicated and differs according to the cellular component. 

With the technology of global SHP2 knockout or “Cre-loxP”-mediated gene conditional knockout in specific cells, the understanding of the role of SHP2 in tissue development becomes more explicit. This article reviews the cell-specific roles of SHP2 in bone homeostasis and discusses the prospective application of the current agonists or inhibitors that target SHP2 in bone-related diseases. Those current findings will provide new opportunities for clinical SHP2-targeting therapy in bone diseases in the future.

## 2. Somatic SHP2 Mutation and Bone Diseases

Somatic SHP2 mutations cause a range of human diseases with developmental disorders. Such patients usually have a normal weight and height at birth, but the developmental deficits become more apparent with age. These characteristics are particularly manifested in NS with SHP2 GOF, which is caused by the *PTPN11* gene mutation. NS patients have growth retardation with a delay of the puberty onset of approximately two years and a deficiency of pubertal growth peak, which leads to a relatively short stature [22]. The bone mineral density is decreased in NS patients [23]. In skeletal deformities, about 15% of NS patients develop thoracic scoliosis. Chest deformity, such as superior pectus carinatum and inferior pectus excavatum, have also been reported [24]. Another developmental disease caused by the SHP2 GOF mutation is LEOPARD syndrome, which results from heterozygous missense mutations in exons 7, 12, and 13 of the *PTPN11* gene. Patients with LEOPARD syndrome also exhibit age-related facial dysmorphisms and skeletal abnormalities [25]. Mandibular osteolytic lesions and chest deformation have been reported in LEOPARD patients [26,27]. SHP2 LOF leads to the development of metachondromatosis, which is a rare genetic disease characterized by osteochondromatosis and enchondromatosis. Patients with metachondromatosis develop exostosis or enchondroma, which in turn affect bone growth [16,28]. Exostosis commonly occurs in the metaphysis of the hands and feet. It can cause a valgus deformity of the finger joints and can regress spontaneously in adolescence for some patients [16,28]. Enchondroma is commonly found in the diaphysis and metaphysis of long bone, which can occupy the normal bone structure, gradually compress the nerves or blood vessels and then lead to nerve palsy or avascular necrosis of the femoral heads [16,28]. The phenotypes of these diseases indicate that somatic SHP2 mutations are closely related with bone development and homeostasis. 

There is insufficient evidence shows that the skeletal deformities caused by SHP2 mutations can progress into malignancies. The probability of carcinogenesis is low in enchondromas [28,29]. However, in other systems, the risk of deterioration in patients with SHP2 mutations is higher than that in the general population. For instance, NS patients have 3.5 times the risk of cancer than the general population [24] and LEOPARD patients have shown an increased risk of neoplasia, particularly in lymphoproliferative disease [27]. Abnormalities of the cardiovascular system, immune system and skin have also been reported [22,25]. In the cardiovascular system, up to 50–60% of NS patients have pulmonary stenosis and 20% have hypertrophic cardiomyopathy [24]. In the immune system, myelomonocytic leukemia and B-cell acute lymphoblastic leukemia have been reported in patients with SHP2 mutations [14,30,31]. These symptoms indicate that SHP2 plays multiple roles in the development of various tissues and organs. 

## 3. SHP2 in Osteoblast Lineage Cells

Bone formation occurs in two forms, intramembranous ossification and cartilaginous ossification. Flat bones, such as the skull, mandible, maxilla and collarbone, develop in the way of intramembranous ossification, where MSCs differentiate into osteoblasts directly. On the other hand, long bones, such as the femur and tibia, form in the way of cartilaginous ossification. Initially, MSCs aggregate and differentiate into chondrocytes that proliferate and secrete a large amount of cartilage matrix. The blood vessels, subsequently, infiltrate in the cartilage matrix. After that, the MSCs surrounding the vessels differentiate into osteocytes and trigger the osteogenesis process. In this part, we summarized the effects and mechanisms of SHP2 in regulating the multiple biological processes of osteoblast lineage cells and concluded the pathological manifestations of SHP2 conditional knockout mice (Table 1).

### 3.1. Mesenchymal Stem Cells

MSCs play an important role in both intramembranous and endochondral ossification. During intramembranous ossification, MSCs differentiate into osteoblasts directly to promote bone growth. In endochondral ossification, MSCs differentiate into early proliferative chondrocytes and, consequently, establish a cartilage growth plate. MSCs, alone or in combination with scaffold materials, have effectively repaired skull or periodontal defects in multiple animal models. Due to the pleotropic differentiation potential, MSCs are considered to be ideal seed cells for tissue regeneration. MSCs transduced with the SHP2 gene promote skull defect repair in rats, accompanied by the upregulation of the expression of osteogenesis-related transcription factors, including osteocalcin (OCN), ALP and RUNX2 [41]. SHP2 inhibition weakens the effect of CGA (chlorogenic acid, a polyphenolic compound extracted from Chinese herb) on MSCs’ proliferation and osteogenic differentiation through PI3K/Akt signaling [42,43]. The abnormal migration of MSCs leads to ectopic bone formation or skeletal dysplasia. SHP2 interacts with P0-related protein (PZR), which is a member of the transmembrane protein to regulate MSCs migration [44]. However, the SHP2 inhibitor (NSC87877) enhanced the proliferation and osteogenic differentiation in stem cells from exfoliated deciduous teeth (SHED) [45]. Taken together, SHP2 participate in intramembrane ossification and endochondral ossification by regulating the proliferation and differentiation of MSCs. 

Prrx1 is a marker of undifferentiated MSCs that can differentiate into chondrocytes and osteoblasts. As one of the osteogenic precursors, Prrx1+ cells possess high regenerative capacities during fracture healing [32,33]. SHP2 are required for Prrx1+ cells to differentiate into osteoblasts. Mice with a deletion of SHP2 in Prrx1-expressing mesenchymal osteochondroprogenitors (SHP2_Prrx1_ KO mice) displayed skeletal dysplasia and impaired ossification in the skull, long bones, ribs, limbs and joint [17,34]. Endochondral ossification and exostoses were also exhibited. Gene array analyses revealed that chondrogenic transcription factors, such as SOX9, Acan, Col2a1, IHH and Col10a1, were increased in SHP2-deficient Prrx1+ cells. Mechanistically, SHP2 promoted the chondrogenic differentiation primarily implicated by SOX9 phosphorylation and SUMOylation mediated by the cAMP-dependent protein kinase (PKA) signaling pathway. The elevation of SOX9 directed osteochondroprogenitors to differentiate toward chondrocytes and delayed their osteogenic differentiation [34]. 

In addition to regulating the function of chondrocytes in endochondral ossification, Wang et al. found that SHP2 also regulated osteoblast differentiation in intramembranous ossification [17]. In SHP2_Prrx1_ KO mice, the skull bone’s ossification was defective due to the poor osteogenic differentiation of MSCs. In vitro osteogenic induction experiments proved that SHP2 deficiency in Prrx1+ cells resulted in a reduced expression of osteogenic transcription factors such as ALP, Col1a1, Ctnnb1, Sp7 and RUNX2. Mechanistically, the impaired osteogenic differentiation was associated with the TGF-β and BMP signaling pathways. SHP2 inhibited TGF-β-evoked SMAD2/3 phosphorylation but promoted BMP2-evoked SMAD1/5/8 phosphorylation in MSCs, indicating that SHP2 was required for the maturation and function of osteoblasts by regulating TGF-β/BMP signaling [17]. In addition, Lapinski et al. [46] found impaired MAPK and AKT signaling were related with poor osteogenic differentiation in SHP2_Prrx1_ KO mice.

### 3.2. Osteoblasts

In osteoblast formation, osteoprogenitors express RUNX2 and differentiate into preosteoblasts. Then, osterix is expressed in preosteoblasts and, finally, defines the cell differentiation to osteoblasts [2]. Osteoblasts secrete matrix proteins, which mineralize into osteoids and embed the osteoblasts during their transition to osteocytes. Bglap is highly expressed in mature osteoblasts, which plays an important role in regulating bone calcium metabolism. Deleting SHP2 in Bglap+ osteoblasts has been shown to inhibit osteogenic differentiation by RUNX2/Osterix7 signaling. Mice lacking SHP2 in their Bglap+ cells (SHP2_Blagp_ KO mice) characterized scoliosis, osteoporosis, osteochondromas and enchondromas. The expression of the terminal osteogenic markers, Dmp1 and Sost, was decreased in Bglap+ cells with SHP2 deletion, leading to impaired osteocyte maturation, matrix production and mineralization function. In addition, local osteoclastogenesis were enhanced due to increased STAT3-mediated RANKL production in SHP2-deficient osteoblasts. Collectively, impaired osteoblast function and enhanced osteoclast activation lead to osteoporosis in SHP2_Blagp_ KO mice. In addition to osteoblasts and osteocytes, a number of chondrocytes also express Bglap, indicating that osteochondromas and enchondromas may be related with SHP2 deletion in Bglap+ chondrocytes [18].

### 3.3. Osteoclast Progenitors/Osteoclasts

Osteoclasts are multinucleated giant cells that are differentiated from the hematopoietic cells of the monocyte-macrophage lineage. When induced by cytokines in the microenvironment, preosteoclast cells in bone marrow will rapidly fuse and differentiate into mature osteoclasts, which can exert the function of bone resorption. Macrophage colony-stimulating factor (M-CSF) and RANKL are two key cytokines in regulating osteoclastogenesis. Upon M-CSF stimulation, SHP2 regulates the proliferation and differentiation of bone marrow macrophage cells (BMMs) into osteoclast precursors through activating AKT signaling [47]. SHP2-deficient BMMs showed decreased proliferation and osteoclastogenesis, and this inhibitory effect was related with the inability of the RAS/ERK signaling pathway [48]. The Nfatc1 is an indispensable transcription factor for osteoclastogenesis and pre-osteoclast fusion. Zhou et al. found that SHP2 promoted osteoclastogenesis by regulating Nfatc1 expression [35]. 

Mice with a deletion of SHP2 in the endogenous lysozyme M (LysM)-labeled osteoclast precursors (SHP2_lysM_ KO mice) developed age-related osteopetrosis, indicating reduced osteoclasts activity [8]. Similarly, in mice with deleted SHP2 in CTSK labeled osteoclasts (SHP2_CTSK_ KO mice), an osteopetrotic phenotype was developed with increased bone density and decreased TRAP+ positive osteoclasts [8,35]. Glycoprotein 130 (GP130) cytokines promoted osteoclastogenesis in vitro. Interestingly, the mice blocking SHP2/RAS/MAPK signaling emanating from GP130 showed an increased osteoclast formation, suggesting that GP130-dependent SHP2 played a role in inhibiting osteoclastogenesis [49]. 

### 3.4. Chondrocytes

SHP2 plays an important role in regulating the proliferation and differentiation of chondrocytes [15,39]. In SHP2 deficiency chondrocyte pellet cultures, the early-hypertrophic chondrocyte transcripts were increased, while the late-stage hypertrophic chondrocytes transcripts were decreased. This suggests that SHP2 depletion maintains chondrocytes at the proliferative stage with a strong proliferative capacity but a poor terminal differentiation capacity. The abnormal terminal differentiation may be a result of activated IHH signaling and inhibited ERK1/2 signaling [15]. SOX9, a major chondrogenic transcription factor, inhibits the β-catenin-mediated osteogenic differentiation. SHP2-deficient chondrocytes showed increased SOX9 and decreased β-catenin expression, which in turn disrupted the transdifferentiation of the chondrocytes [38]. The hyperactivation of RAS/ERK signaling also occurred in SHP2 mutant chondrocytes and resulted in decreased ALP activity, demonstrating the important role of SHP2 on endochondral ossification [36]. Shao et al. [39] found that SHP2-deficient chondrocytes had a higher expression of BMP6 and a stronger osteoinductive capacity through promoted Smad1/5 expression within MSCs. The aberrant chondrocytes regulated ectopic new bone formation and aggravated ankylosing spondylitis progression. In addition, SHP2 mediated the chondrocyte degeneration caused by interleukin 1β (IL-1β), with an increased expression of matrix-metalloproteinase 3 (MMP3) and matrix-metalloproteinase 13 (MMP13), as well as a decreased expression of Col2a1 [50]. Taken together, SHP2 not only leads to the proliferation and differentiation of abnormal chondrocytes, but also affects osteogenesis through the paracrine pathway. 

Col2a1 is a marker of proliferative chondrocytes. Kim et al. found severe spinal deformities were developed in SHP2_col2a1_ KO mice, characterized by scoliosis, kyphosis and lordosis; an increased growth plate cartilage in the vertebral body was also observed [36]. Bowen et al. [15] found an expansion and disturbance of the growth plates, as well as enchondroma and exostosis formation in SHP2_col2a1_ KO mice, and this phenotype may be related with ERK1/2 signaling or SOX9-mediated β-catenin signaling. Meanwhile, they found that deleting SHP2 in Fsp-expressing fibroblasts also induced exostosis, suggesting that SHP2-deficient cells could induce the chondrogenesis of normal cells by paracrine signals. In addition to the altered morphology and structure of the spine mentioned above, SHP2 deficiency also caused enchondromas and ectopic cartilage nodules on the hands, feet and ribs in mice. An increased expression of IHH and fibroblast growth factor 2 (FGF2) were observed in aberrant cartilage nodules, while MAPK activation was impaired [16], indicating that IHH, FGF2 and MAPK may regulate metachondromatosis progression in SHP2_col2a1_ KO mice. In addition, Kamiya et al. [37] found that a severe deformity in the mandibular condyle was developed in SHP2_col2a1_ KO mice, and abnormal cartilaginous nodules appeared in the subchondral bone and trabecular bone. This implied that maxillofacial bone development might also be regulated by SHP2. Interestingly, no significant metachondromatosis development was observed, whilst a reduction in the bone mineral density was observed, in mice with ablated SHP2 in Col10α-1 hypertrophic chondrocytes [38]. The distinctive skeletal phenotypes when deleting SHP2 in proliferating or hypertrophic chondrocytes revealed that SHP2 played a stage-specific effect in chondrogenesis and skeletal metabolism. 

Shao et al. [39,40] found that SHP2 deficiency in a population of chondrocytes expressing CD4 also contributed to skeletal malformation. In SHP2_CD4_ KO mice, bone fusion, joint stiffness and osteophytes owing to abnormal endochondral ossification were observed in the spine, wrist, sacroiliac, hip and knee joints. This provides a new experimental animal model for ankylosing spondylitis (AS) and a promising intervention target for drug development. The targeted inhibition of aberrant chondrocytes significantly alleviated the symptoms of ectopic ossification and bone fusion in SHP2_CD4_ KO mice. Mechanistically, SHP2-deficient chondrocytes promoted the ectopic osteogenesis of bone MSCs by activating BMP6/Smad1/5 signaling. Meanwhile, increased bone resorption caused by osteoclasts was also developed in SHP2_CD4_ KO mice.

CTSK is primarily expressed in osteoclasts. Interestingly, SHP2_CTSK_ KO mice exhibited scoliosis, exostoses and enchondromas similar to metachondromatosis [8]. Lineage tracing revealed the role of CTSK-expressing cells as a novel chondroid progenitor population, which explained why SHP2 knockdown in CTSK-expressing cells led to abnormal cartilage development. Furthermore, FGFR and ERK were decreased, while IHH and Pthrp expression were increased, in SHP2-deficient chondroprogenitors. The inhibition of IHH significantly reduced exostoses formation in SHP2_CTSK_ KO mice. This revealed that SHP2 inhibited metachondromatosis by activating FGFR/ERK to combat excessive IHH and Pthrp production in chondrocytes. SHP2’s enzyme activity also increased within chondrocytes in OA patients and in an experimental murine model [51]. Overall, SHP2 plays an important role in maintaining cartilage homeostasis and development. Targeting SHP2 can be an effective therapeutic strategy in treating cartilaginous diseases.

## 4. SHP2 in Bone Niche Cells

The role of the niche microenvironment in bone remodeling has become increasingly concerning. The immune cells, vasculature ECs, as well as the sensory and sympathetic nerves, in bone tissue help to maintain bone homeostasis by providing precursor cells with a specific microenvironment and integrating biological signals [2]. We next reviewed the role of SHP2 in regulating niche cells to indirectly modulate bone remodeling.

### 4.1. Immune Cells

Both the innate and adaptive immune systems are closely linked with bone development. In innate immune systems, macrophages and dendritic cells can differentiate into osteoclasts to participate in bone remodeling directly and can also secrete cytokines to influence bone remodeling indirectly [52,53]. Macrophages are actively involved in both bone resorption and bone formation as osteoclasts originate from mononuclear macrophages. In the early stage of inflammation, macrophages polarize to M1-type macrophages and secrete proinflammatory cytokines, such as interleukin 6 (IL-6) and tumor necrotic factor-α (TNF-α), to promote bone resorption [54]. During the alleviating phase of inflammation, M2-type macrophages remove tissue debris and promote the osteogenic differentiation of MSCs [55,56,57]. Meanwhile, interleukin 4 (IL-4) and interleukin 6 (IL-10) cytokines secreted by M2-type macrophages inhibit bone resorption [58]. SHP2 negatively regulates the activation of NOD-like receptor thermal protein domain associated protein 3 (NLRP3) in macrophages, thus inhibiting the secretion of proinflammatory cytokines IL-1β and interleukin 18 (IL-18) [19]. The ablation of SHP2 in macrophages evocates the production of T-cell chemoattractant, C-X-C motif chemokine 9 (CXCL9), and mediates the recruitment of the effector T cell [59]. Furthermore, SHP2 could also regulate macrophage proliferation and polarization to M2-type macrophages [48,60]. Recently, a new type of bone macrophages, referred to as osteomacs, was discovered on the surface of bone cortex. They are a group of cells featured with F4/80, CD68, Mac-3, CD169 positive and TRAP negative, and promoted bone formation by secreting BMP2 and BMP6 [61,62]. However, studies on the effect of SHP2 on osteomacs are absent and still need to be explored. Dendritic cells (DCs) can differentiate into osteoclasts at the early developmental stages. Osteoclasts differentiated from DCs produce higher IL-1β than steady-stage osteoclasts, further promoting bone resorption [63]. SHP2 is essential for the migration of DCs. Xu et al. found that SHP2 deficiency inhibited DC migration via the Rho/MLC/cofilin/Pyk2 signaling pathway [64].

In adaptive immune systems, T cells and B cells are activated by antigen presenting cells and exert an immunoregulatory role in bone. B cells are associated with bone resorption via modulating the RANKL/(osteoprotegerin) OPG ratio in the bone marrow microenvironment. RANKL promotes osteoclastogenesis, while OPG acts to the opposite effect. In physiological conditions, B cells are the main cells producing OPG and account for 64% of the total marrow OPG production [65,66]. Osteoporosis was developed in B cell- deficient mice [67]. SHP2 has been reported to promote the proliferation of pre-B cells [68,69], however, the underlying mechanisms are still under investigation.

The T cells, upon parathyroid hormone (PTH) stimulation, would promote the activity of osteoclasts through an increased RANKL/OPG ratio. Simultaneously, they differentiate into TH17 and secrete IL-17a, inducing more RANKL production from osteoblasts to promote osteoclastogenesis [67]. Furthermore, regulatory T cells inhibit bone resorption through the secretion of IL-4, IL-10 and TGF-β, and induce osteoclast apoptosis through activating the intracellular NIK/IDO pathway by direct contact with osteoclasts [70]. SHP2 negatively regulates T-cell activation and immune function. Programmed death 1 (PD-1) is mainly expressed in activated T-cells. As a downstream of PD-1, SHP2, when recruited by PD-1, suppresses T-cell activation by inhibiting the PI3K/AKT signaling pathway [71,72]. Based on this evidence, the effect of SHP2 on bone development and disease by regulating immune cells should not be underestimated. However, the role of SHP2 on the above immune cells is mostly studied in tumor or inflammation diseases, and regrettably few studies have been performed in bone immune cells, which remains an issue to be further investigated in the future (Figure 1). 

### 4.2. Vasculature Endothelial Cells 

Blood vessels transport oxygen and nutrients to the bones and excrete metabolic waste, which play an important role in bone development and defect repair. In recent years, targets or drugs that promote angiogenesis in the treatment of osteoporosis and fractures achieved good efficacy in animal models [73,74]. Vascular ECs in bone tissue can be divided into the H type with, high CD31 expression, and the L type, with low CD31 expression. H-type blood vessels are surrounded by progenitor cells secreting type I collagen, while L-type blood vessels are surrounded by hematopoietic stem cells, with almost no bone progenitor cells around [75]. The number of H-type vascular ECs decreases significantly with age, which showed the same trend as the decrease in bone mass. A significant increase in bone mass and H-type vascular ECs were observed after treatment with anti-osteoporosis drugs or H-type vascular activators. Mechanistically, H-type vascular ECs promote the proliferation and osteogenic differentiation of their surrounding precursor osteocytes through the high expression of hypoxia-Inducible Factor 1-α (HIF-1α), platelet derived growth factor (PDGFA), TGFβ1, TGFβ3, fibroblast growth factor (FGF1) and Noggin [76,77]. However, in pathological conditions, such as in OA, H-type vessels increase in subchondral bone and release MMP-9 or RANKL, leading to osteoclast formation and cartilage matrix degradation [78]. These results indicate that targeting vascular ECs may provide a new therapeutic approach for bone formation and repair.

SHP2 regulates vessel formation and maintains vascular homeostasis. SHP2 deletion or inhibition disrupt the endothelial barrier integrity and result in embryonic hemorrhage and lethality [79]. In addition, the downregulation of SHP2 reduces vascularization at the skin wound healing site [80]. HIF-1α is an important transcription factor under hypoxia; SHP2 promotes vascularization by stabilizing and aggregating HIF-1α [20]. In addition, SHP2 prevents endothelial apoptosis through the MAPK and AKT pathways [8]. Recently, the targeted inhibition of SHP2 has become a novel tumor therapeutic strategy. The SHP2 inhibitor SHP099 disturbs tumor neovascularization by promoting STAT3 phosphorylation and inhibiting ERK1/2 phosphorylation [81]. SHP2 promotes the expression of proangiogenic factor SOX7 through c-JUN signaling to promote pathological angiogenesis in tumors, while a decreased tumor angiogenesis was observed in mice with SHP2-knockdown endothelial cells [82]. As the role of SHP2 is different under different physiological and pathological states, SHP2 agonists or inhibitors should be rationally applied in different disease models (Figure 2).

### 4.3. Nervous System

The role of the central nervous system (CNS) and peripheral nervous system on bone remodeling has become a widely-discussed topic. In the CNS, numerous neurogenic factors expressed in hypothalamic have been reported to regulate bone remodeling, such as leptin, serotonin, semaphorins, neuropeptide Y, etc. [83,84,85]. Xia et al. found that hippocampal neurons could release small extracellular vesicles enriched with miR-328a-3P and miR-150-5P to promote osteoblast function after craniocerebral injury and accelerate fracture healing [83]. For the peripheral nervous system, sensory and sympathetic nerves distribute richly in bone marrow or periosteum, secreting multiple neurotransmitters that regulate bone formation and resorption [86,87,88]. Calcitonin gene-related peptide (CGRP), a neuropeptide, functions as a pro-osteogenetic cytokine in the regions with active osteogenesis, such as epiphysis and periosteum [89]. Li et al. found that CGRP-positive nerve fibers increased in the bone fracture region and regulated osteoblasts to promote fracture healing [90]. Other neuropeptides, such as substance P (SP), vasoactive intestinal peptide (VIP), and nerve growth factor (NGF), also have an important role for bone reconstitution [91,92,93]. 

SHP2 is required for proliferation, self-renewal, migration and the differentiation of neural stem cells (NSCs) and Schwann cells [13,94,95]. As a multipotent cell, NCCs migrate and differentiate into various cell lineages, which further develop into craniofacial tissues, cartilage, smooth muscle, neurons and glia. When deleting SHP2 in NCCs, the mice displayed severe craniofacial defects [21]. SHP2 directly bind to the leptin receptor and activate downstream ERK signaling. Mice deleting SHP2 in neurons showed enhanced leptin resistance [96,97]. In addition, mitochondrial dysfunction in neurons is closely related with the development of Parkinson’s disease. SHP2 activation enhanced mitochondrial autophagy and thus alleviated Parkinson-like symptoms in mice [98]. Unfortunately, few studies have reported the relationship between SHP2 and the neurotransmitters associated with bone remodeling. Further studies focusing on the interaction between the skeletal and nervous system will have important implications for bone disease recovery (Figure 3).

## 5. Prospects of SHP2 Agonists and Inhibitors in Treating Bone-Related Diseases

To date, several SHP2-targeted agonists and inhibitors have been reported to function in skeletal disorders. Lovastatin is a classical drug for hypercholesterolemia and significantly lowers low-density lipoprotein and triglyceride levels [99]. Recently, Liu et al. found that lovastatin was an agonist of SHP2 [98] and effectively promoted fracture healing [100,101]. The most common side effects of lovastatin are gastrointestinal symptoms, such as flatulence and dyspepsia [102]. Although lovastatin is an activator of SHP2, there is insufficient clinical data to validate its carcinogenetic effect. Instead, lovastatin has been reported to treat cancers, such as glioblastomas, breast, liver, cervical, lung and colon cancer, in recent years [99,103]. SHP099, an oral SHP2 inhibitor, suppressed the progression of various malignant tumors, such as colon cancer, neuroblastoma, lung cancer and acute myeloid leukemia (AML) [104,105,106,107]. Recently, its role in bone or joint diseases has been brought into sight. Liu et al. found that an intraperitoneal injection of SHP099 alleviated osteoarthritis progression, as SHP099 inhibited DOK1-mediated uridine phosphorylase 1 (UPP1) expression and subsequently upregulated the uridine level to maintain cartilage homeostasis [51]. In addition, the injection of SHP099-loaded thermal hydrogel had a protective effect on the articular disc in the intervertebral disc degeneration rat model [108]. NSC-87877, another inhibitor of SHP2, was reported to block the formation and fusion of osteoclasts, inhibit bone resorption, and reverse the bone loss in osteoporosis mice [35,109]. As an optimization of SHP099, TNO155 is the first SHP2 inhibitor to enter clinical studies [110]. In animal experiments, the main adverse effects are phototoxicity and phospholipidosis. To date, a negative effect on bone tissue has not been reported [110]. Other highlighted SHP2 inhibitors, such as NSC-17199, tautomycetin and cryptotanshinone, all showed specific and inhibitory effects [111]. However, further studies of their effect on bone diseases are anticipated.

As SHP2 regulates multiple downstream signaling pathways, such as MAPK, PI3k/AKT and IHH signaling, targeting SHP2-mediated signaling may provide a potential therapeutic approach to bone development and bone-related diseases. Tajan et al. [112] reported that treatment with U0126, an inhibitor of ERK1/2, improved the impaired growth plate in SHP2 KO mice. In addition, SHP2 deficient chondrocytes showed increased Pthrp expression evoked by IHH signaling. The IHH inhibitor, PF-04449913, ameliorated metachondromatosis in mice with a lack of SHP2 in CTSK labeled chondroid progenitors [8]. 

In conclusion, the role of SHP2 modulators in non-malignant skeletal disorders is multi-directional. Fracture healing was promoted by lovastatin [100,101]. However, in osteoporosis, osteoarthritis and intervertebral disc degeneration, the application of the SHP2 inhibitors, SHP099 and NSC-87877, was reported to alleviate disease progression. This may be related to the effector cells regulated by SHP2 and should be studied further to clarify its therapeutic mechanism. The combined application of SHP2-targeting drugs (lovastatin or current SHP2 inhibitors, JAB-3068, RLY-1971, RMC-30, TNO155, BBP-398) and bone-modifying agents (BMAs) has not been reported for the treatment of bone-related diseases. Future studies may be interesting in the development of new strategies. 

## 6. Conclusions

SHP2, a classic member of the protein tyrosine phosphatases family, has a significant impact on skeletal development and homeostasis maintenance. The role of SHP2 on bone remodeling is comprehensive and multi-targeted. Firstly, SHP2 participate in bone development by regulating the proliferation and differentiation fate of osteoblast lineage cells and osteoclasts. Under pathological conditions, abnormal SHP2 expression exacerbates disease progression. Secondly, SHP2 has regulatory effects on the immune cells, ECs and nervous systems associated with bone development. Therefore, targeting SHP2 is a promising therapeutic strategy in bone related diseases. However, the different roles of SHP2 in various cell types and physiopathological conditions need to be considered comprehensively. Meanwhile, how to target specific cell populations in order to prevent the adverse effects on other cells still needs to be explored.

## Figures and Tables

**Figure 1 ijms-24-02202-f001:**
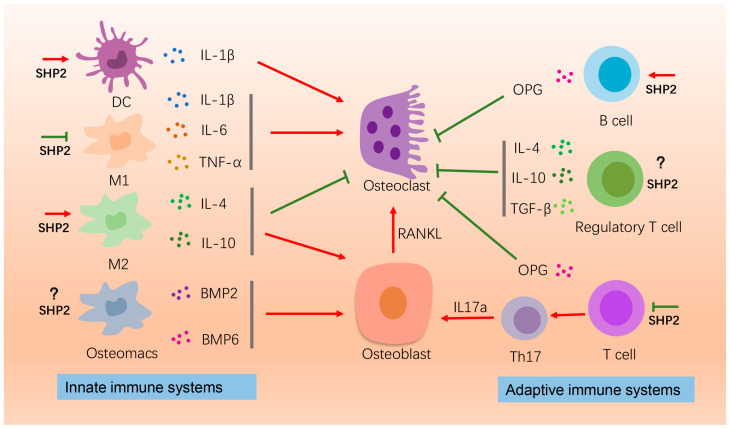
The effects of SHP2 on bone homeostasis by affecting immune cells. Here the roles of macrophages, B-cells, and T-cells on osteogenesis or osteoclatogenesis are depicted. IL-1β, IL-6, TNF-α released by the M1 macrophages promote osteoclastogenesis for bone resorption. SHP2 suppresses this process. The M2 macrophages secrete IL-4 and IL-10 to promote bone formation and suppress bone resorption. SHP2 enhances this process. Osteomacs produce BMP2 and BMP6 to promote bone formation. The effect of SHP2 on osteomacs is unknown. B-cells produce OPG to inhibit osteoclastogenesis and SHP2 promotes pre-B cells proliferation. Regulatory T cell releases IL-4, IL-10 and TNF-α to inhibit osteoclastogenesis. T cells, on the one hand, produce OPG to inhibit osteoclasts, and on the other hand differentiate into TH17, promoting osteoblast activity by producing IL-17α. This process is inhibited by SHP2. “Red arrow” refers to promotion; “Green arrow” refers to inhibition.

**Figure 2 ijms-24-02202-f002:**
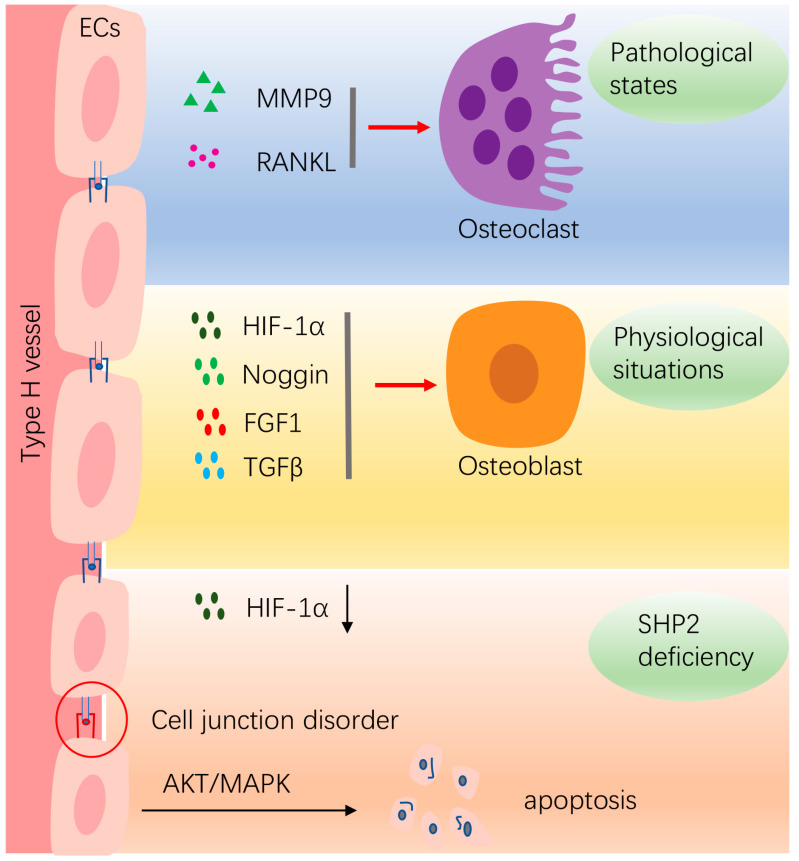
The effects of SHP2 on bone homeostasis by affecting vasculature ECs. In physiological situations, ECs release multiple cytokines, such as HIF-1α, Noggin, FGF1, TGFβ, etc., to promote osteogenesis. In pathological conditions, such as in inflammatory conditions, ECs secrete MMP9 and RANKL to promote osteoclastogenesis. Deletion of SHP2 in ECs decreases HIF-1α expression. Furthermore, the junctions between ECs are disordered and the integrity of the vascular barrier is disrupted. The SHP2-deficient ECs undergo apoptosis through activation of MAPK and AKT pathways. “Red arrow” refers to promotion.

**Figure 3 ijms-24-02202-f003:**
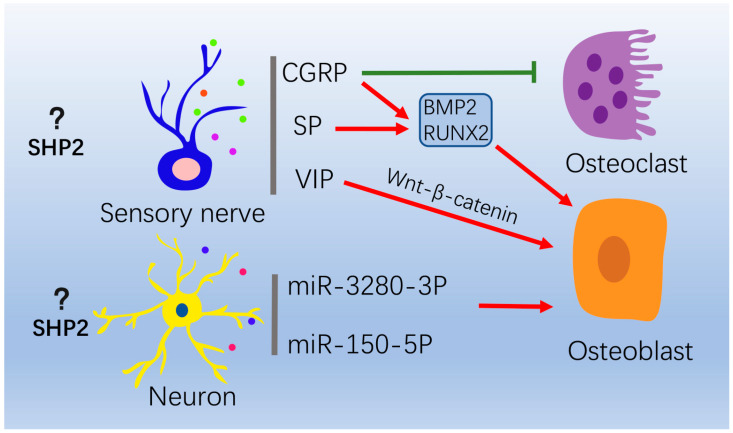
The role of the nervous system on osteoblasts and osteoclasts. In peripheral nervous system, sensory and sympathetic nerves secrete neurotransmitters to regulate bone formation and resorption. CGRP suppresses osteoclasts formation. CGRP and SP promotes osteogenesis via increasing BMP2 and RUNX2 expression. VIP promotes osteogenesis through WNT/β-catenin signaling. In central nervous system, neurons can release extracellular vesicles enriched with miR-328a-3P and miR-150-5P to promote osteogenesis. “Red arrow” refers to promotion; “Green arrow” refers to inhibition.

**Table 1 ijms-24-02202-t001:** Mouse models of SHP2 deletion with skeletal abnormalities.

Target Cell	Phenotype	Mechanism	Related Signaling Pathways	References
Prrx1+ mesenchyme stem cells	Skeletal dysplasia; impaired ossification in skull, long bones, ribs, limbs and joint; pectus excavatum and pectus carinatum; endochondral ossification; exostoses	Chondrogenic transcription factors: SOX9, Acan, Col2a1, Col10a1↑ Osteogenic transcription factors: ALP, Col1a1, Ctnnb1, Sp7, RUNX2↓	TGF-β/SMAD2/3, BMP2/SMAD1/5/8↓	[17,32,33,34]
Bglap+ osteoblasts	Scoliosis, osteoporosis, osteochondromas, enchondromas	Osteogenic transcription factors: RUNX2, Osterix7, Dmp1, Sost↓	STAT3/RANKL↑	[18]
LysM+ osteoclasts precursors	Age-related osteopetrosis	Osteoclastogenesis transcription factors: Nfatc1↓	AKT↓, RAS /ERK↓	[8]
CTSK+ osteoclasts	Osteopetrosis, scoliosis, exostoses and enchondromas	Reducing osteoclasts activity	MAPK↓, IHH↑	[8,35]
Col2a1+ chondrocytes	Spinal deformities, scoliosis, kyphosis, lordosis, enchondroma and exostosis	Chondrogenic transcription factors: SOX9, BMP6↑ Osteogenic transcription factors: ALP↓	IHH↑, MAPK↓, β-catenin↓	[15,16,36,37]
Fsp1+ expressing fibroblasts	Exostosis	Inducing normal cells undergo chondrogenesis by paracrine	Unknow	[15]
Col10a-1+ chondrocytes	Bone mineral density reduction	Chondrogenic transcription factors: SOX9↑ Osteogenic transcription factors: Ibsp, RUNX2, Ctnnb1↓	WNT/β-catenin↓	[38]
CD4+ chondrocytes	Bone fusion and joint stiffness, ankylosing spondylitis, osteoporosis	Chondrogenic transcription factors: Col2a1, Col10a1, Acan, and Pthrp↑ Osteogenic transcription factors: RUNX2, Sp7, Ocn↑	BMP6/Smad1/5↑, ERK1/2↓ IHH↓	[39,40]

↑, up regulation; ↓, down regulation. Abbreviations: SOX9, SRY-Box transcription factor 9; Acan, Aggrecan; Col2a1, alpha-1 type II collagen; Col10a1, alpha-1 type X collagen; ALP, Alkaline phosphatase; Col1a1, alpha-1 type I collagen; Ctnnb1, Catenin beta 1; Sp7, osterix; Dmp1, Dentin matrix acidic phosphoprotein 1; Sost, Sclerostion; Nfatc1, Nuclear factor of activated T cell cytoplasmic 1; Ibsp, Integrin binding sialoprotein; Pthrp, Parathyroid hormone-related protein; Ocn, Osteocalcin; Prrx1, Paired related homeobox 1; Bglap, Bone gamma-carboxyglutamate protein; LysM, lysozyme MFsp1, Fibroblast-specific protein 1; CTSK, Cathepsin K.

## Data Availability

Not applicable.

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
