# Peer review of "The Cell-Specific Role of SHP2 in Regulating Bone Homeostasis and Regeneration Niches"

_ijms, 2023, doi:10.3390/ijms24032202_

Round 1

Reviewer 1 Report

Overall, excellent summary of SHP2 pathway in bone disease. table and figures are organized and illustrated nicely - although formatting of table could be improved (for example, there are several overhangs that make the visual presentation a bit hard to read). also the paper has several minor grammar/spelling issues 

section 2:

- Line 76: please provide a bit more detailed description of the syndromes caused by SHP2 GOF/LOF mutations mentioned - example if they are associated with abnormal bone production or more of a osteoporotic phenotype. Do they have increased risk of any malignancies? please provide citation for last sentence (line 92-93) and brief description of the abnormalities seen in CV/immune systems. 

- section 5: could benefit from expanding this section and discuss in more detail the clinical implications of SHP2 targeting or agonists. any obvious toxicities from the drugs (particularly any conner for acceleration of malignancies from the SHP2 agonists)? for the non-malignant skeletal disorders, is the focus on SHP2 agonism or antagonism? are there studies (or rationale to support) looking at combination treatments of SHP2 modulators and existing BMAs like bisphosphonates, RANK-L inhibitors, sclerostin antibodies?

Author Response

Response to Reviewer 1 Comments

Point 1: Overall, excellent summary of SHP2 pathway in bone disease. table and figures are organized and illustrated nicely - although formatting of table could be improved (for example, there are several overhangs that make the visual presentation a bit hard to read). also the paper has several minor grammar/spelling issues.

Response 1: Thanks for the revierwer’s suggestion.

We have adjusted the format of table 1 (page4, line123).

We checked up the whole manuscript and corrected several grammar/spelling issues.

‘Exhibits’ is replaced with ‘exhibit’ (page2, line87)

‘Initially, MSCs aggregate and differentiate into chondrocytes. With the proliferation of chondrocytes and secretion of cartilage matrix, blood vessels gradually infiltrate and surrounding MSCs differentiate into osteocytes.’is replaced with ‘Initially, MSCs aggregate and differentiate into chondrocytes that proliferate and secrete a large amount of cartilage matrix. The blood vessels, subsequently, infiltrate in the cartilage matrix. After that, the MSCs surrounding the vessels differentiate into osteocytes and trigger the osteogenesis process.’(page3, line116)

Delete extra ‘transmembrane protein’ (page5, line142)

‘Together’ is replaced with ‘Taken together’ (page5, line145)

‘osteogenesis’ is replaced with ‘osteoblast’ (page5, line150)

‘preosteoblast’ is replaced with ‘preosteoblasts’ (page5, line173)

‘osteoblast’ is replaced with ‘osteoblasts’ (page5, line175)

‘osteoclast generation’ is replaced with ‘osteoclastogenesis’ (page6, line198)

‘osteoclasts differentiation’ is replaced with ‘osteoclastogenesis’ (page6, line201)

‘osteoclast’ is replaced with ‘osteoclasts’ (page6, line206)

‘promote’ is replaced with ‘promoted’ (page6, line207)

‘cartilage’ is replaced with ‘chondrocytes’ (page6, line229)

‘osteoclastogetic activity’ is replaced with ‘the activity of osteoclasts’ (page8, line316)

‘SHP2 is required for proliferative, self-renewal, migration and differentiation of neural stem cells (NSCs) and Schwann cells’ is replaced with ‘SHP2 is required for proliferation, self-renewal, migration and differentiation of neural stem cells (NSCs) and Schwann cells.’ (page10, line397)

Point 2: - Line 76: please provide a bit more detailed description of the syndromes caused by SHP2 GOF/LOF mutations mentioned - example if they are associated with abnormal bone production or more of a osteoporotic phenotype.

Response 2: Thanks for the reviewer’s advice. We have added this part as follows:

Page 2, line 81, ‘The bone mineral density is decreased in NS patients [23]. In skeletal deformities, about 15% of NS patients develop thoracic scoliosis. Chest deformity such as superior pectus carinatum and inferior pectus excavatum have also been reported [24].’

Page2, line 88, ‘Mandibular osteolytic lesions and chest deformation have been reported in LEOPARD patients [26, 27].’

Page3, line 92-97, ‘The exostosis commonly occurs in metaphysis of hands and feet. It can cause a valgus deformity of the joint of fingers and regress spontaneously in adolescence of some patients [16, 28]. Enchondroma is commonly found in diaphysis and metaphysis of long bone, which can occupy the normal bone structure, gradually compress the nerves or blood vessels and then lead to nerve palsy or avascular necrosis of femoral heads [16, 28].’

Point 3: Do they have increased risk of any malignancies?

Response 3: Thanks for your question. Most of the skeletal deformities caused by SHP2 mutations are not life-threatening to patient. However, the risk of deterioration in other systems resulting from SHP2 mutations is higher than that in the general population. We added this point in Line as follows:

Page3, line 99-105, ‘There is insufficient evidence that shows the skeletal deformities caused by SHP2 mutations can progress into malignancies. The probability of carcinogenesis is low in enchondromas [28, 29]. However, in other systems, the risk of deterioration in patients with SHP2 mutations is higher than that in the general population. For instance, NS patients have 3.5 times the risk of suffering cancer than the general population [24] and LEOPARD patients showed an increased risk of neoplasia, especially in lymphoproliferative disease [27].’

Point 4: Please provide citation for last sentence (line 92-93) and brief description of the abnormalities seen in CV/immune systems.

Response 4: Thanks for your kind remind and suggestion. We added references citations in Page3, line108. The abnormalities of cardiovascular system, immune system and skin have also been reported [22, 25].  

We also added a brife description of the abnormalities in CV/immune systems.

Page3, Line 106-110, ‘In cardiovascular system, up to 50–60% NS patients have pulmonary stenosis, 20% have hypertrophic cardiomyopathy [24]. In immune system, myelomonocytic leukemia and B-cell acute lymphoblastic leukemia have been reported in patients with SHP2 mutations [14, 30, 31]. These symptoms indicate that SHP2 plays multiple roles in the development of various tissues and organs.’

Point 5: Could benefit from expanding this section and discuss in more detail the clinical implications of SHP2 targeting or agonists. any obvious toxicities from the drugs (particularly any conner for acceleration of malignancies from the SHP2 agonists)?

Response 5: Thank you for the constructive advice. Bone metastasis is common in malignant tumors, causing active bone resoption and accompanied pain. SHP2 inhibitors may surpress such osteoclastogenesis thus alleviating the tumor-related bone damage. Despite lack of relevant reports, this is a meaningful study and requires further validation. We added this point as follows:

Page11, Line 418-425, ‘Currently, several SHP2-targeted agonists and inhibitors have been reported to function in skeletal disorders. Lovastatin is a classical drug for hypercholesterolemia and significantly lowers low-density lipoprotein and triglyceride levels [99]. Recently, Liu et al. found that lovastatin was an agonist of SHP2 [98] and effectively promoted fracture healing [100,101]. The most common side effects of lovastatin are gastrointestinal symptoms, such as flatulence and dyspepsia [102]. Although lovastatin is an activator of SHP2, there is no sufficient clinical data to validate its oncogenesis. Instead, lovastatin has been reported to treat cancers such as glioblastomas, breast, liver, cervical, lung, and colon cancer in recent years [99,103].’

Page 11, Line 436-438, ‘As an optimization of SHP099, TNO155 is the first SHP2 inhibitor to enter clinical studies [110]. In animal experiments, the main adverse effects are phototoxicity and phospholipidosis. So far, negative effect on bone tissue has not been reported [110]. ‘

Point 6: For the non-malignant skeletal disorders, is the focus on SHP2 agonism or antagonism?

Response 6: Thanks for your question. We added this point as follows:

Page12, Line 450-455, ‘In conclusion, the role of SHP2 modulators in non-malignant skeletal disorders is multi-directional. Fracture healing was promoted by lovastatin [100, 101]. However, in osteoporosis, osteoarthritis and intervertebral disc degeneration, the application of SHP2 inhibitors SHP099 and NSC-87877 was reported to alleviate disease progression. This may be related with the effector cells regulated by SHP2 and should be studied further to clarify its therapeutic mechanism.’

Point 7: Are there studies (or rationale to support) looking at combination treatments of SHP2 modulators and existing BMAs like bisphosphonates, RANK-L inhibitors, sclerostin antibodies?

Response 7: Thanks for your question. We added this point as follows:

Page12, 455-458, ‘Combined application of SHP2-targeting drugs (lovastatin or current SHP2 inhibitors, JAB-3068, RLY-1971, RMC-30, TNO155, BBP-398) and bone-modifying agents (BMAs) has not been reported in treating bone-related diseases. Future studies may be interesting in the development of new strategies.’

Reviewer 2 Report

Authors reviewed participation of SHP2 in bone related diseases and its mechanisms. The influence of abnormal expression or mutations of SHP2 on osteogenic, chondrogenic and niche cells were summarized.  Current agonists or inhibitors targeting SHP2 in bone-related diseases were also described. This review is helpful for researchers and clinicians.

P2, lines 52 – 68 “Although extensive studies have…….” -------à The sentences are overlapped in other pages consider abbribation.

Author Response

Response to Reviewer 2 Comments

Point 1: Authors reviewed participation of SHP2 in bone related diseases and its mechanisms. The influence of abnormal expression or mutations of SHP2 on osteogenic, chondrogenic and niche cells were summarized.  Current agonists or inhibitors targeting SHP2 in bone-related diseases were also described. This review is helpful for researchers and clinicians.

#1 P2, lines 52 – 68 “Although extensive studies have…….” -------à The sentences are overlapped in other pages consider abbribation.

Response 1: Thanks for the reviewer’s advice. We have abbreviated this content as follows:

Page2, line 60-66, ‘Knockdown of SHP2 in osteochondroprogenitors cells or osteoblasts resulted in decreased osteogenic differentiation and bone formation capacity [17, 18]. Deletion of SHP2 in osteoclasts could impair osteoclasts formation [8]. In addition to these stromal cells which directly involved in bone metabolism, other niche cells in bone microenvironment could also be regulated by SHP2. Inhibition of SHP2 in macrophages and vascular ECs reduced the secretion of proinflammatory cytokines [19] and impaired angiogenesis, separately [20].’

Reviewer 3 Report

This review article is so organized and provides fine information to the readers.

However, to improve this article, I would like to suggest only one point.

1.       I think this article gives little information on SHP2.

Therefore, the authors should describe more detailed general information on SHP2: for example, the target protein/genes, the function including cell differentiation in other cells, or the regulatory gene/protein.

Please add such information in the INTRODUCTION.

Author Response

Response to Reviewer 3 Comments

Point 1: This review article is so organized and provides fine information to the readers. However, to improve this article, I would like to suggest only one point. I think this article gives little information on SHP2. Therefore, the authors should describe more detailed general information on SHP2: for example, the target protein/genes, the function including cell differentiation in other cells, or the regulatory gene/protein.

Please add such information in the INTRODUCTION.

Response 1: Thanks for the reviewer’s advice. We added the information as follows:

Page2, Line 51-57,’ In MAPK signaling pathway, hyperactivation of ERK1/2 by SHP2 mutations leads to growth retardation in NS patients [14]. Reduced ERK1/2 phosphorylation by SHP2 deletion results in delayed chondrocytes terminal differentiation and maturation [15]. Meanwhile, SHP2 negatively regulates IHH protein and AKT in mouse metachondromatosis model [16]. Moreover, SHP2 participates in osteoblasts maturation and differentiation by promoting BMP2 and RUNX2/OSTERIX signaling [17], while inhibits osteoclastogenesis by STAT3-mediated RANKL production [18].’